# The Internal Structure of the WISC-V in Chile: Exploratory and Confirmatory Factor Analyses of the 15 Subtests

**DOI:** 10.3390/jintelligence12110105

**Published:** 2024-10-25

**Authors:** Marcela Rodríguez-Cancino, Andrés Concha-Salgado

**Affiliations:** Department of Psychology, Universidad de La Frontera, Temuco 4811322, Chile; andres.concha@ufrontera.cl

**Keywords:** intelligence, assessment, factor structure, children, age group

## Abstract

The WISC-V is a widely used scale in clinical and educational settings in Chile. Given that its use guides critical decision-making for children and adolescents, it is essential to have evidence of its psychometric properties, including validity based on internal structure. This study analyzed the factor structure of the WISC-V through an exploratory (EFA) and confirmatory (CFA) approach considering the age range of 853 children and adolescents between 6 and 16 years. We obtained evidence favoring the four-factor structure in the EFA, which is a clearer organization in the 15–16 age group. In the confirmatory stage, the best four- and five-factor models showed factor loadings greater than 0.4, except for one subtest in the processing speed domain in the 6–8 age group. The internal consistency ranged from acceptable to good estimates for the best two models. The results support the use of hierarchical factor structures of four and five factors, which offer specific advantages and disadvantages discussed in the article. The implications of these findings in both the professional area of psychology and future research are discussed.

## 1. Introduction

Professionals who use educational or psychological tests must ensure that the assessments they perform with them have a sound scientific basis that supports the conclusions and decisions that emerge from their administration ([50]). In the clinical or educational context, where decisions relevant to the support of children and adolescents must be made, it is essential to have rigorous information on the psychometric properties of the most commonly used tests to ensure that their use is appropriate, ethical, and fair ([2]; [14]; [27]; [39]; [40]; [53]).

In this regard, the Standards for Educational and Psychological Testing ([1]), indicate that it is crucial to show evidence of validity when examining the psychometric properties of a test, since it demonstrates the extent to which evidence and theory support its intended application.

Internal structure validation is a crucial aspect of determining the validity of a test. It offers insights into the construct being tested, its functioning, and its structure ([35]). Examining the consistency between the items and the underlying dimensions of the test helps determine if the proposed interpretation of its scores is supported ([36]; [50]).

The evaluation of dimensionality (factor structure) is a way of gathering evidence about its internal structure. This process entails comparing the hypothesized factor structures with those acquired by applying suitable confirmatory statistical methods (e.g., confirmatory factor analysis). The results obtained through such an analysis must then be discussed in terms of their consistency with the theory underlying the test ([5]; [26]; [49]).

In the clinical or educational setting, one of the most widely used tests worldwide is the Weschler Intelligence Scale for Children (WISC), developed in the United States and adapted to different cultural settings in Europe and Latin America ([21]; [31]; [37]; [42]; [48]). For more than 75 years of research, this scale has demonstrated its clinical utility for a variety of purposes, whether in the identification of an intellectual disability, specific learning disorders, clinical interventions, or neuropsychological assessment ([3]; [7]; [16]; [17]; [23]; [56]).

Since its creation in 1949, the scale (see Table 1) has undergone a series of revisions and adjustments to accurately represent and accommodate the cultural and technological advancements that have occurred with each successive generation ([42]; [48]; [56]; [60]). The WISC-V is the latest available version of this scale, and it includes improvements in its theoretical foundations, psychometric properties, clinical utility, and administration formats ([10]; [48]; [56]).

### 1.1. Factor Structure of the WISC-V

The factor structure of the WISC is one of the aspects that has been modified based on these revisions, and given that it underpins the interpretation of its scores, its psychometric exploration is of great importance to ensure its use in different contexts ([15]). According to the Technical and Interpretative Manual of the fifth American edition of the scale ([56]), and as seen in Table 1, the internal structure of WISC has progressed from the understanding of a general intellectual ability model (FSIQ) composed only of a verbal IQ (VIQ) and a performance IQ (PIQ) to hierarchical factor models that include a general factor and four or five cognitive domains.

Decisions about the structure of the WISC-V were based on neurocognitive research, neurodevelopmental theories, and structural intelligence models ([21]; [30]). According to [56] ([56]), there is now broad agreement on hierarchical intelligence models, which identify a general intelligence factor at the top and several broad, related, but distinguishable skills at a lower level. Although there are different models, many agree that verbal comprehension, visuospatial reasoning, fluid reasoning, working memory, and processing speed should be included as essential components. According to [20] ([20]), [43] ([43]), and [61] ([61]), the structure of the WISC-V is closely aligned with the Cattell, Horn, Carroll framework (CHC), which describes a three-stratum model of intelligence, with the general intelligence factor at the top (second-order factor like the FSIQ), and broad and narrow abilities (first-order factors like the primary index scales) and more specific skills at the bottom.

The latest edition of the scale (WISC-V) included changes compared to the structure of the previous version (WISC-IV), which had a hierarchical factorial configuration with a second-order factor called general intelligence (FSIQ) or *g*, and four first-order factors called verbal comprehension (VC), perceptual reasoning (PR), working memory (WM), and processing speed (PS). To enhance the understanding and interpretation of the measures, and based on factor analysis and theoretical foundations, the fifth version splits PR into two indices, namely visuospatial reasoning and fluid reasoning. It also included a task that measures visual working memory, especially useful when it is necessary to differentiate it from verbal auditory working memory when evaluating different clinical conditions ([31]; [48]; [56]; [60]).

Thus, the factor structure of the WISC-V includes a second-order factor called general intelligence (FSIQ) and five first-order factors called verbal comprehension (VC), visuospatial (VS), fluid reasoning (FR), working memory (WM) and processing speed (PS), configuring a hierarchical five-factor intelligence model ([20]; [21]; [31]; [42]; [43]; [48]; [55], [56]; [61]).

According to [61] ([61]), these five indicators on the WISC-V are equivalent to the broad abilities proposed in the CHC model in the following way: VC corresponds to comprehension knowledge (*Gc*), VS to visual processing (*Gv*), FR to fluid reasoning (*Gf*), WM to working memory capacity (*Gwm*), and PS to processing speed (*Gs*).

This factor structure has been reported in several countries, such as France, ([58]), Spain ([57]), Canada ([55]), Taiwan ([12]), and Germany ([59]), as proposed by [56] ([56]) in the original US version. It should be noted that in a recent study, [61] ([61]) explored the equivalence of the WISC-V five-factor model with the French, Spanish, and US standardization samples and found that the five-factor hierarchical model demonstrated an excellent fit in all three samples independently. In addition, these authors demonstrated strict factorial invariance between France, Spain, and the United States, supporting the generalizability of the constructs across populations speaking different languages.

### 1.2. Criticisms of the Factor Structure Proposed for the WISC-V

Despite the clinical, psychometric, and theoretical foundations of the WISC-V’s internal structure, it has faced criticism from multiple independent authors (not linked to the publisher) who have performed exploratory factor analyses (EFA), confirmatory factor analyses (CFA), and bifactor analyses (BFA) with data from the standardization samples of their respective countries ([9]; [8]; [18]; [33]; [54]).

[8] ([8]) explored the internal structure of the WISC-V in the US standardization sample (*n* = 2200). The results of the EFA highlight the clear presence of four factors (VC, VS, WM, and PS), without a fifth latent factor (such as FR) as proposed by [56] ([56]). On the other hand, the authors mention that the CFA results with models that included five first-order factors are inadmissible because they present negative variances in FR, which is an improper solution. In contrast, four-factor models that merge VS and FR in a single factor showed a better fit.

These same results were found in some studies that used standardization samples, namely [54] ([54]; 880 Canadian children) and [9] ([9]; 415 United Kingdom children), who examined the factor structure for the 16 subtests of the WISC-V and found no support for the five-factor model proposed by [56] ([56]). The same conclusion was reached by [33] ([33]), who replicated the analysis on the French standardization sample (*n* = 1049) using the 15 WISC-V subtests. A similar result was found in Spain, using data from a standardized sample comprising 1008 examinees throughout the country ([18]).

Likewise, analyses of the internal structure of the WISC-V conducted on large clinical samples, by [11] ([11]) with 2512 participants and [15] ([15]) with 5359, show that the hierarchical factor models composed of four first-order factors have a better fit than the five-factor model proposed by [56] ([56]), replicating the structure of the previous version of this scale (WISC-IV).

A recent study, which used a new strategy for analyzing the internal structure of a test known as the exploratory graph analysis (EGA), explored the dimensionality of the WISC-V in the French standardization sample, suggesting the presence of three dimensions that coincide with the cognitive domains of processing speed (PS), verbal comprehension (VC), and perceptual reasoning (PR), discarding the distinction between VS and FR ([34]).

The studies discussed thus far add inconclusive evidence about the number of dimensions in the instrument. Views have recently emerged favoring structures other than the five-factor structure, such as four or three dimensions. Another source of contention is related to the hierarchical nature of the construct. Several independent authors propose that a bifactor model for the WISC-V offers a better factor solution than hierarchical (or second-order) models since *g* results from a direct measurement of the subtests, making it more parsimonious or simplified. In hierarchical models, *g* is an indirect measure or an “abstraction of abstractions” ([11]), where the five primary indices are an unnecessary intermediate layer between *g* and the subtests. These authors emphasize that clinical interpretation should rest solely on the FSIQ ([8]; [43]; [60]).

In contrast to this idea, [60] ([60]) argue for using the hierarchical five-factor model proposed by [56] ([56]) for the WISC-V. They support their argument with statistical evidence, such as the adequate fit indicators of the second-order model and the minimal improvement in fit seen with bifactor models. They also provide theoretical support, noting that interpretations of scores are based on an internationally validated theoretical model for intelligence-CHC tests, which is not consistent with bifactor models. This viewpoint is also shared by [32] ([32]). Additionally, [60] ([60]) emphasize clinical arguments, highlighting the consistency of relationships observed in clinical practice between measured skills and the proposed indices.

The results presented highlight the need to study different variants of the models, whether five-factor or four-factor, or hierarchical, bifactor, or oblique, to analyze the advantages and disadvantages of one over the other comparatively.

### 1.3. Factor Structure of the WISC-V in Chile

The WISC-V was standardized in Chile with a sample of 754 children and adolescents ([48]). Its psychometric properties include an internal consistency that varies between acceptable (0.645) and excellent (0.941) reliability values for its subtests and excellent (between 0.900 and 0.968) for all its indices (see Table 2).

On the other hand, it has evidence of validity based on its internal structure and association with different variables (WISC-III and WAIS-IV). Regarding the internal structure, the scale exhibits satisfactory fit indices, enabling us to confidently assert that the hierarchical five-factor model presented by [56] ([56]) for the US sample, both 10 and 15 subtests, are adequately replicated for the Chilean population ([48]). It should be noted that in Chile, the internal structure of the WISC-V has also been explored in a rural sample, finding an adequate level of fit for the five-factor model of 15 and 10 subtests and for the model of 7 primary subtests that make up the FSIQ ([46]).

Furthermore, in Chile, evidence supports the measure’s equivalence by testing the factorial invariance of the WISC-V based on urban/rural origin. This analysis revealed partial metric invariance, indicating discrepancies in analogies that may indicate the existence of measurement bias in this particular subtest ([45]).

In a more recent study, [44] ([44]) tested the invariance of two WISC-V factor models (hierarchical and oblique) in the standardization sample (*n* = 740) according to the gender and age group of their participants (6–8, 9–11, 12–14, 15–16). The results showed complete invariance according to sex but incomplete according to age group due to discrepancies in the subtests that are part of the fluid reasoning index (matrix reasoning and figure weights). This suggests that the items on these subtests do not measure these skills in the same way in young children as in adolescents. Based on this, considering that the equivalence of the measure according to the participants’ age was not demonstrated, the authors note that, in addition to exploring the nature of the possible measurement bias, it would be advisable to test other factor models (e.g., four-factor models), to verify whether there are better factorial structures, or different and more relevant models within each of the age ranges studied, an aspect that has not yet been studied in Chile.

Considering the myriad international evidence and the questions raised by independent authors regarding the internal structure of the WISC-V in the different cultural contexts in which it has been translated, adapted, and standardized, the present study seeks to generate psychometric evidence that guarantees its use is appropriate and fair for children and adolescents of different ages in the Chilean population.

It is important to note that when translating a test, there will always be discrepancies between the original and translated versions. Therefore, it cannot be assumed that an adapted version accurately measures the intended constructs or automatically captures the expected relationships between these proposed constructs in the same way in different groups ([37]; [52]). This must be examined and demonstrated.

On the other hand, explaining whether the best internal configuration of the WISC-V consists of four or five factors or if it differs depending on whether the respondent is a child or an adolescent is especially critical for professionals who frequently use this instrument and must make sense of its scores ([37]), as this will guide the decision-making process that will directly impact the life and academic trajectory of those being assessed.

### 1.4. The Present Study

To provide information on these issues, the present study examined the latent factor structure of the 15 primary subtests of the Chilean version of the WISC-V by (1) conducting an exploratory factor analysis (EFA) on the total sample and by age group, (2) performing a confirmatory factor analysis (CFA) on the total sample, and (3) comparing the best-fitted models by age group.

## 2. Materials and Methods

### 2.1. Participants

The total sample included 853 participants aged 6–16 years (*M*_age_ = 10.9, *DS*_age_ = 3.039), with 693 (81%) from urban and 160 (19%) from rural areas.

The children in the urban sample correspond to secondary data obtained from the Standardization Project of the WISC-V Scale in Chile. The data collection was carried out by the Center for the Development of Inclusion Technologies research team at the Pontificia Universidad Católica de Chile (CEDETi-UC) through a purposive sampling method based on sex (balanced) and the region of the country and a stratified method depending on whether the school was public, private, or mixed administered as a proxy socio-economic status ([48]). The rural sample corresponds to primary data collected by the research team through non-probability purposive sampling.

In both samples, the inclusion criteria were as follows: (a) aged between 6 and 16 years and 11 months, and (b) not recently assessed on a similar scale. The exclusion criteria were a clinical diagnosis and permanent or temporary special educational needs. Detailed information on frequencies by gender and age group is given in Table 3. The age groups for this research were defined based on the previous study by [44] ([44]).

### 2.2. Instruments

#### Wechsler Intelligence Scale for Children, Fifth Edition (WISC-V)

An individually administered clinical instrument assesses the cognitive functioning of children and adolescents ([47]; [56]). The Chilean version of this scale, available since 2018, presents adequate psychometric properties in this population, with appropriate levels of reliability (see Table 2), evidence of internal structure validity (in urban and rural samples), and evidence of its relationship with other variables ([46]; [48]).

The Chilean version of the WISC-V includes 15 subtests, ten primary and five complementary, organized into five cognitive domains (see Table 2). The scores of the subtests are expressed as scale scores (*M* = 10, *SD* = 03). With *M* = 100 and *SD* = 15, the FSIQ and the indices are expressed as composite scores ([31]; [47]; [56]).

### 2.3. Procedure

In urban and rural samples, WISC-V was administered only to children or adolescents previously authorized by their parents, who agreed to participate voluntarily. Each child was assessed individually at their schools and during regular class hours by research assistants who had passed a training course in applying and correcting the WISC-V. The duration of each application was between 60 and 90 min, depending on the child’s performance.

Regarding ethical considerations, all the procedures carried out for the data collection of the urban sample were approved by the Scientific Ethics Committee in Social Sciences, Arts, and Humanities of the Pontificia Universidad Católica de Chile. The Universidad de La Frontera Scientific Ethics Committee approved the research protocol for the rural sample. The informed consent documents provided detailed information to parents, children, and adolescents about the project’s objectives, the administration process, and the right to withdraw from participation at any time without negative consequences for the participants. These documents also detailed confidentiality safeguards, indicating that the data would only be used anonymously and for scientific and academic purposes.

To utilize the data from the urban sample in this study, permission was obtained from CEDETi-UC, the institution holding the rights to the WISC-V in Chile, which also endorsed this study.

### 2.4. Data Analyses

The scale scores of the 15 subtests of the Chilean WISC-V standardization sample served as the basis for the exploratory factor analyses (EFA) and confirmatory factor analyses (CFA) conducted for the study.

#### 2.4.1. Exploratory Factor Analyses

First, we wanted to identify the number and composition of common factors (latent variables) necessary to explain the common variance of the 15 indicators analyzed. EFA were performed using the principal-axis factoring method and oblique rotation (Oblimin) with a Pearson correlation matrix in JASP 0.18.3 ([28]). Horn’s parallel analysis (HPA) determined the number of factors in the total sample and age subgroups.

#### 2.4.2. Confirmatory Factor Analyses

Factor models were then tested with various configurations replicating the CFA reported in the American WISC-V Technical and Interpretative Manual ([56]). Through decisions guided by theory and previous empirical evidence, these models tested different allocations of the subtests to various factors, as follows:M_1_ =a model where all subtests load directly on a general ability factor as the only indicator responsible for the intercorrelations between subtests.M_2_ =a “traditional” two-factor Wechsler model (VIQ and PIQ) that distinguishes verbal from performance, present in initial versions of the instrument.M_3_ =a three-factor model combining verbal comprehension with auditory working memory on the one hand; fluid reasoning, visuospatial ability, and visual working memory on the other; and a third factor containing only processing speed.M_4_ =four models of four factors each, where VC and PS (with the same subtests) are common to all four variants but with the following distinctions in the other two factors that comprise them: M_4a_,as in the WISC-IV, which includes a reasoning factor (PR) that merges fluid and visuospatial reasoning and another for working memory (WM);M_4b_,which, based on findings in cognitive neuroscience, combines fluid reasoning and working memory into a single factor since they share a common function of the prefrontal cortex (FR+WM) and another of visuospatial skills (VS);M_4c_,the same as M_4a_ but with cross-loadings from the arithmetic subtest in WM and PR; andM_4d_,the same as M_4a,_ but with cross-loadings from the arithmetic subtest in WM, PR, and VC.M_5_ =five five-factor models, including VC, VS, FR, WM, and PS, which differ only in the location of the arithmetic subtest loading(s):M_5a_,only in WM (as in the four-factor models);M_5b_,only in FR;M_5c_,in WM and FR;M_5d_,in WM and VC; andM_5e_,in WM, FR, and VC, as proposed by [56] ([56]).

CFA were implemented using the robust maximum likelihood estimator (MLR) in Mplus 8.10 ([41]). In terms of model fit, the Tucker–Lewis index (TLI), comparative fit index (CFI), and root mean square error of approximation (RMSEA) were used. According to [6] ([6]) and [24] ([24]), an optimal fit was defined as TLI and CFI > 0.95 and RMSEA < 0.05, as well as a reasonable fit when the TLI and CFI > 0.90 and RMSEA < 0.08.

#### 2.4.3. Age-Group Model Comparison

To determine the best fit for the four- and five-factor hierarchical models according to age group (6–8, 9–11, 12–14, and 15–16), we considered AIC and BIC indices (lower values indicate better fit) and the chi-square difference test using the Satorra-Bentler scaled chi-square from the Mplus software ([51]). We also took the model’s theoretical soundness into account. The selected best options were analyzed within the four age groups using CFA.

#### 2.4.4. Reliability

Internal consistency was calculated through the omega coefficient at a subscale level (Ω_S_) and at a general factor level (Ω_GF_) using the information from the CFA in JASP.

## 3. Results

### 3.1. Exploratory Approach

In the total sample, the EFA with the 15 primary and complementary WISC-V subtests generates a distinguishable and theoretically coherent four-factor solution (VC, PR, WM, and PS), with good fit indices and adequate factor weights, which replicates the model of its predecessor, WISC-IV.

Within the four age groups, the four-factor organization remains consistent with a good fit, factor loadings, and theoretical interpretability, although with minor variations in cross-loadings and the size of factor weights. Cross-loadings > 0.3 were observed for IN and CA in G1, FW in G2, and AR in G3. Likewise, factor loadings < 0.4 were identified in FW in G1, G2, and G3, MR in G2 and G3, CA in G1, and CO in G2. Interestingly, in G4, unlike the other three groups, a clear organization of the subtests with factor weights greater than 0.4 and no cross-loadings was observed (see Table 4).

### 3.2. Confirmatory Approach

The CFA in the total sample made it possible to identify some models with good fit indices and factor weights, although others had poor indicators. M_1_, M_2_, and M_3_ had a poor fit. The four four-factor models (M_4_) and the five five-factor models (M_5_) showed adequate fit indices. However, M_4a_ and M_5a_ had comparative advantages by showing good fit indices, chi-square differences with other models, theoretical consistency with the CHC model, and no cross-loadings. Notably, M_4a_ corresponds exactly to the four-factor model provided by the EFA (see Table 5).

### 3.3. Best Fitted Models Comparison by Age Group

Once the best fit and theoretical soundness models were identified, they were analyzed in the total sample (see Figure 1) and then within each age group (see Table 6) to verify their stability. Both M_4a_ and M_5a_ had adequate fit indices in all four age subgroups. All the factor loadings were greater than 0.4 in all models and age groups, except for CA in PS in G1 (6–8) in M_4a_ and M_5a_. This characteristic disappears in the other age groups in both models, with the weight of CA increasing as age increases.

Regarding the magnitude of the factor loadings in the cognitive domains (first-order factors), it should be noted that in G1 and G2 in M_4a_ and G2 in M_5a_, the WM domain presents loadings greater than 0.9. Something similar occurs with FR in G2 and G3 in M_5a_. This situation could represent a possible redundancy with *g* in these subgroups, an issue that tends to balance out in the older age group, since the domain weights in *g* are more balanced.

### 3.4. Internal Consistency

Reliability estimates in the M_4a_, at the subscale level (Ω_S_), were acceptable to good, except for PS, which was questionable, although the overall factor level (Ω_GF_) was acceptable. In M_5a_, reliabilities were good in only two of the five first-order factors, although it was good at the general factor level.

## 4. Discussion

In Chile, the use of the WISC-V is mandatory in the educational context ([38]). Therefore, examining its psychometric properties and generating evidence of its reliability and validity are essential to guarantee the quality of the assessments made with this scale. This study aimed to provide evidence of validity by examining the internal structure of the test on the basis that it is possible to find support at the international level for the model proposed by [56] ([56]), as well as addressing concerns raised by independent authors.

The first objective was to explore the factor structure of the WISC-V using an EFA. The 15 subtests were naturally grouped into four theoretically interpretable factors (VC, PR, WM, and PS) in the total sample, without the presence of a fifth factor (such as FR), which is consistent with reports from other countries using exploratory analyses ([9]; [8]; [18]; [33]; [54]). These authors note that the WISC-V is over-factored since there is no empirical evidence of the existence of a fifth factor.

Similarly, grouping the subtests into the four factors found in the total sample is maintained in all four groups. However, the factor solutions in G1, G2, and G3 show problems such as cross-loadings greater than 0.3 (some theoretically inconsistent, such as CA in VC) and weak factor loadings (e.g., FW weights 0.250 on F2 in G1), i.e., lower than what is conventionally accepted as adequate. It is important to point out that the factor solution in the older age group (15–16) is similar to that of the total sample: parsimonious and without problems that could cast doubt on its interpretability.

The EFA findings show cross-loadings in G1, G2, and G3, which naturally disappear in G4. Therefore, the results of this study, in relation to the structure of intelligence, could represent a pattern like the one proposed by the differentiation hypothesis ([4]; [29]; [62]). This hypothesis states that cognitive skills are undifferentiated at the beginning of the life cycle and gradually break down into more specific or specialized skills. Corroborating this proposition according to age or ability level with other methodological approaches would be an interesting line of future research.

To achieve the study’s second objective, various confirmatory factor models were tested, and the results show that the hierarchical structures of four and five factors present comparatively better-fit indices. The models of one, two, and three factors were discarded. This empirical evidence supports the plausibility of a four-factor structure and the five-factor one originally proposed by [56] ([56]) in the United States. The results of the five-factor model coincide with those reported in France, Spain, Canada, Taiwan, and Germany ([12]; [55], [57], [58], [59]) indicating that the intelligence construct, measured with the WISC-V in the Chilean sample, is composed of a set of positively correlated latent abilities and structured hierarchically, coinciding with the proposal of the CHC model ([20]; [43]; [61]).

This study also sought to determine whether a four- or five-factor model shows a more relevant fit according to the age of the participants. To this end, the “best models” were chosen and tested according to age range. They were identified based on psychometric aspects (review of the fit indices and factor loadings) and theoretical aspects (agreement with the proposals of the CHC model). In addition, considering the concerns of the independent authors, we chose models that did not include improper solutions or cross-loadings, placing AR only in the working memory domain.

Psychometrically, comparisons of the “best models” of four and five factors in the Chilean sample suggest that for all age ranges, both configurations fulfill the current requirements shared by the scientific community that indicate when a model presents a plausible factor solution ([6]; [24]), with minimal differences in their fit indices. Both structures have issues related to weak factor weights, which can indicate a lack of relevance of the subtest in its intended domain. Conversely, there may be very high domain loadings that independent authors would find questionable, as they could be empirically redundant with *g* ([9]).

Regarding the latter, the CFA shows empirical redundancies in WM for G1 and G2 in the four-factor model and G2 in the five-factor model. This is also observed in FR, for G2 in the four-factor model, and for G3 in the five-factor model. No empirical redundancies were found in either model in the older age group (15–16). In this regard, it should be noted that evidence has shown a close relationship (sometimes indistinguishable) between WM and intelligence or executive functions, identifying it as a central process of cognition and a good predictor of academic performance in children with typical and atypical development ([13]; [22]; [25]). A close relationship between WM and FR has also been observed in neuroimaging studies that have shown that solving WM and FR tasks requires the activation of the same region of the prefrontal cortex ([13]).

The empirical redundancy found in this study could be explained by this close interrelation, observing that WM is a central component of intelligence in younger children, which in middle childhood coexists with FR. Then, in the following age range, abstract reasoning (FR) becomes the central component until, in adolescence, the set of cognitive domains, differentiated, make up the general intellectual ability, without the role of one standing out over the other. It should be noted that in adolescence, FR reaches its full development ([13]).

These results are particularly interesting for the Chilean sample, as they suggest that the evaluation of WM is central to understanding their general cognitive functioning in younger children. Moreover, [22] ([22]) report that there is evidence showing that WM training can enhance fluid intelligence in children aged 7–11. Therefore, further exploration of these results in the Chilean sample, along with new lines of research on the relationship between WM components and general intelligence, could be a significant practical contribution to assessing and intervening cognitive processes in childhood.

At the subtest level, it is interesting that in the CFA of the four-factor model and the five-factor model, the CA subtest exhibits the same trend of gradually increasing its factor loading as the age of the subjects increases. In addition, the weights of CA are practically identical in both configurations, as follows: weak in the youngest children, medium in the intermediate ages, and highest in the last age range. Effective task resolution on the CA subtest depends on autonomous, quick, and successful decision-making and strategy selection, as well as the ability to inhibit impulsive responses ([31]). This result could indicate that in this subtest, the performance of the Chilean sample reflected the progressive development of these skills, which is explained by the maturational process that occurs during childhood and adolescence. This provides further proof that the role of CA in determining information processing speed grows increasingly important as individuals age, making its interpretation more reliable during adolescence. In the other VP subtests and the verbal comprehension and working memory dimensions, there is no gradual increase with age, as observed in CA. However, it is notable that the factor loadings are very similar in the four- and five-factor models. This suggests that there are no noteworthy differences in structuring these domains, regardless of whether a four- or five-factor configuration is considered.

In terms of determining whether the best internal configuration of the WISC-V for the Chilean sample is composed of four or five factors, the empirical evidence of this study shows arguments in favor of both models (fit indices, theoretical consistency) but also against (models with cross-loadings or empirical redundancies). One advantage of the four-factor model is its more evenly distributed configuration of subtests, with three domains comprising four subtests and one composed of three subtests. This contrasts with the five-factor model, which has two factors with two subtests, one with three subtests and two with four subtests, as illustrated in Figure 1. On the other hand, the five-factor model has a better internal consistency of the general intelligence construct and is perfectly aligned with the CHC theoretical proposal ([61]), which is internationally recognized as a solid foundation for interpreting the scores on current intelligence tests. As mentioned above, this was one of [56] ([56]) arguments for retaining the five-factor model.

The four-factor structure of the WISC-IV emerged after several attempts to integrate theory and research, incorporating the CHC theoretical model of intelligence in the interpretation of its scores for the first time ([19]). For the WISC-V, new subtests or combinations of these were incorporated to have increasingly more specific indices or to provide “purer” measures of the constructs evaluated, incorporating VS and FR as two separate factors to increase their practical usefulness ([21]).

[60] ([60]) note that in clinical practice, there is consistency between the skills measured and the indices that make up the five-factor model, where the separation of VS and FR increases the practical usefulness of this version of the scale, as the measurement of nonverbal reasoning skills is more precise. For their part, [43] ([43]) and [32] ([32]) performed a series of CFA to determine the feasibility of separating VS and FR in the WISC-V and found that it was indeed psychometrically admissible. This represents an advantage of five-factor models over four-factor models.

In summary, in the Chilean sample, the four- and five-factor models are defensible at the psychometric level, presenting minimal differences in fit according to age range and similar problems. In theoretical terms, both models are based on the same theoretical framework (CHC), although the five-factor model is more closely aligned. Regarding clinical utility, the five-factor models achieve greater accuracy in the measurement of nonverbal reasoning skills because they have the separation of VS and FR. Considering all these psychometric, theoretical, and practical utility aspects, the results of this study provide evidence to support the use of the five-factor model to interpret WISC-V scores in Chile in children and adolescents.

### Limitations and Future Research

Despite the theoretical and empirical contributions of the present study, its findings cannot be generalized to minority groups of the Chilean population or clinical groups. Given that the use of this scale is widespread in Chile to assess all children who require it, it would be advisable to explore the psychometric evidence in these groups. In addition, it is suggested that the reported findings be investigated with factorial invariance analyses.

## 5. Conclusions

This groundbreaking study in Chile investigated the significance of factor models with various configurations for the WISC-V in a substantial sample of children from different backgrounds across four age groups. Although varied, the results lend support to using the five-factor model proposed by [56] ([56]). Hopefully, these findings will enhance Chilean psychologists’ clinical and educational practice and contribute to fair and relevant evaluations.

## Figures and Tables

**Figure 1 jintelligence-12-00105-f001:**
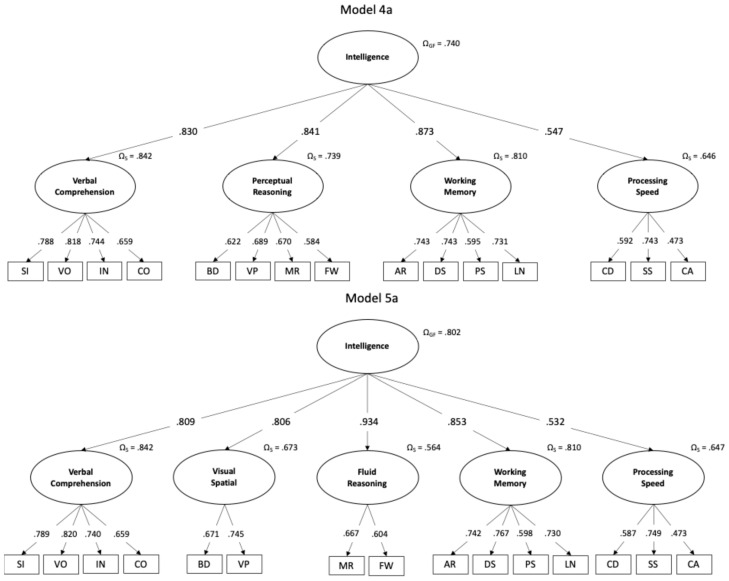
CFAs of the best-fitted models in the total sample.

**Table 1 jintelligence-12-00105-t001:** Versions of the WISC-V since its creation.

Edition	Ages	Subtests	g Level	Indexes Level				
WISC (1949)	5–15	12	Full Scale IQ (FSIQ)	Verbal IQ (VIQ)		Performance IQ (PIQ)		
WISC-R (1974)	6–16	12	Full Scale IQ (FSIQ)	Verbal IQ (VIQ)		Performance IQ (PIQ)		
WISC-III (1991)	6–16	13	Full Scale IQ (FSIQ)	Verbal IQ (VIQ)		Performance IQ (PIQ)		
				Verbal comprehension (VCI)	Freedom from distractibility (FDI)	Perceptual organization (POI)		Processing speed (PSI)
WISC-IV (2003)	6–16	15	Full Scale IQ (FSIQ)	Verbal comprehension (VCI)	Working memory (WMI)	Perceptual reasoning (PRI)		Processing speed (PSI)
WISC-V (2014)	6–16	21	Full Scale IQ (FSIQ)	Verbal comprehension (VCI)	Working memory (WMI)	Fluid reasoning (FRI)	Visual spatial (VSI)	Processing speed (PSI)

*Note:* Table created by the authors.

**Table 2 jintelligence-12-00105-t002:** The classification of subtests according to the cognitive domain and reliability coefficients for the Chilean version of the WISC-V.

Type of Subtest	Cognitive Domain				
	Verbal Comprehension (α = 0.943)	Visual Spatial (α = 0.912)	Fluid Reasoning (α = 0.945)	Working Memory (α = 0.933)	Processing Speed (α = 0.900)
Primary subtest	Similarities (SI; α = 0.921)	Block Design (BD; α = 0.824)	Matrix Reasoning (MR; α = 0.900)	Digit Span (DS; α = 0.907)	Coding (CD; α = 0.898)
	Vocabulary (VO; α = 0.888)	Visual Puzzles (VP; α = 0.903)	Figure Weights (FW; α = 0.941)	Picture Span (PS; α = 0.891)	Symbol Search (SS; α = 0.822)
Complementary subtest	Information (IN; α = 0.910)		Arithmetic (AR; α = 0.900)	Letter-Number Sequencing (LN; α = 0.895)	Cancellation (CA; α = 0.645)
	Comprehension (CO; α = 0.876)				

*Note:* Table created by the authors; α = Cronbach’s alpha.

**Table 3 jintelligence-12-00105-t003:** Characteristics of the sample according to sex and age group.

Age Group (in Years)	Boys		Girls		Missing		Total Sample	
	*f*	%	*f*	%	*f*	%	*f*	%
6–8	112	50.5%	110	49.5%	0	0.0%	222	26.0%
9–11	126	47.4%	139	52.2%	1	0.4%	266	31.2%
12–14	110	48.7%	115	50.9%	1	0.4%	226	26.5%
15–16	64	46.0%	75	54.0%	0	0.0%	139	16.3%
Total sex	412	48.3%	439	51.5%	2	0.2%	853	100%

*Note:* Table created by the authors.

**Table 4 jintelligence-12-00105-t004:** EFAs in the total sample and disaggregated by age group.

	Total Sample				G1 (6–8)				G2 (9–11)				G3 (12–14)				G4 (15–16)			
Sub-Tests	F1	F2	F3	F4	F1	F2	F3	F4	F1	F2	F3	F4	F1	F2	F3	F4	F1	F2	F3	F4
SI	.688				.440				.743				.747				.771			
VO	.840				.719				.715				.860				.884			
IN	.653				.402		.425		.701				.719				.648			
CO	.701				.381				.796				.750				.727			
BD		.585				.539				.490				.668				.621		
VP		.756				.676				.740				.743				.749		
MR		.518				.692				.374				.350				.597		
FW		.334				.250				.316	.298			.312				.575		
AR			.539				.712				.582		.319		.435				.586	
DS			.786				.749				.756				.782				.855	
PS			.432				.317				.450				.608				.469	
LN			.739				.740				.648				.823				.614	
CD				.481				.564				.497				.460				.435
SS				.766				.583				.812				.748				.795
CA				.505	.315			.347				.540				.477				.613
Fit	χ2 = 81.854, df = 51, *p* = .004 CFI = .993, TLI = 0.986 RMSEA [90% CI] = .027 [.015 .037] SRMR = .015 BIC = −262.333				χ2 = 60.431, df = 51, *p* = .172 CFI = .990, TLI = 0.980 RMSEA [90% CI] = .029 [.00 .054] SRMR = .027 BIC = −215.105				χ2 = 48.823, df = 51, *p* = .561 CFI = 1.000, TLI = 1.003 RMSEA [90% CI] = .000 [.000 .037] SRMR = .020 BIC = −235.936				χ2 = 73.401, df = 51, *p* = .022 CFI = .981, TLI = 0.961 RMSEA [90% CI] = .044 [.018 .066] SRMR = .024 BIC = −203.046				χ2 = 70.209, df = 51, *p* = .038 CFI = .979, TLI = 0.957 RMSEA [90% CI] = .052 [.013 .080] SRMR = .029 BIC = −181.450			

*Note:* Factor loadings smaller than 0.25 were omitted; F1 is consistent with the verbal comprehension domain, F2 with perceptual reasoning, F3 with working memory, and F4 with processing speed.

**Table 5 jintelligence-12-00105-t005:** Factorial configurations, goodness-of-fit indexes, and model comparisons of the CFA models in the total sample.

	Factors					Fit Indices									Model Comparison			
Model	F1	F2	F3	F4	F5	χ2	df	*p*	CFI	TLI	RMSEA [90%CI]	SRMR	AIC	BIC	Comparison	Δχ^2^	Δdf	*p*
M_1_	All 15 subtests	-	-	-	-	848.424	90	<.001	.819	0.789	.099 [.093 .106]	.064	59,503	59,717	-	-	-	-
M_2_	**V** SI VO IN CO AR DS LN	**P** BD VP MR FW PS CD SS CA	-	-	-	724.117	89	<.001	.848	0.821	.091 [.085 .098]	.060	59,372	59,590	-	-	-	-
M_3_	**V** SI VO IN CO AR DS LN	**P** BD VP MR FW PS	**PS** CD SS CA	-	--	523.522	87	<.001	.896	0.874	.077 [.070 .083]	.049	59,171	59,399	-	-	-	-
M_4a_ *	**VC** SI VO IN CO	**PR** BD VP MR FW	**WM** AR DS PS LN	**PS** CD SS CA	-	214.571	86	<.001	.969	0.963	.042 [.035 .049]	.034	58,860	59,092	M_4a_ vs. M_4c_ M_4a_ vs. M_4d_ M_4a_ vs. M_5a_ M_4a_ vs. M_5c_ M_4a_ vs. M_5d_ M_4a_ vs. M_5e_	7.497 17.124 4.773 7.402 14.857 16.474	1 2 1 2 2 3	.011 <.001 .026 .029 .001 .001
M_4b_	**VC** SI VO IN CO	**VS** BD VP	**FR+WM** MR FW AR DS PS LN	**PS** CD SS CA	-	283.097	86	<.001	.953	0.943	.052 [.045 .059]	.038	58,931	59,163	M_4b_ vs. M_4c_ M_4b_ vs. M_4d_ M_4b_ vs. M_5a_ M_4b_ vs. M_5c_ M_4b_ vs. M_5d_ M_4b_ vs. M_5e_	76.023 85.650 63.753 75.928 83.343 85.000	1 2 1 2 2 3	<.001 <.001 <.001 <.001 <.001 <.001
M_4c_	**VC** SI VO IN CO	**PR** BD VP MR FW AR	**WM** AR DS PS LN	**PS** CD SS CA	-	207.074	85	<.001	.971	0.964	.041 [.034 .048]	.033	58,853	59,091	M_4c_ vs. M_4d_ M_4c_ vs. M_5c_ M_4c_ vs. M_5d_ M_4c_ vs. M_5e_	9.627 0.095 7.360 8.977	1 1 1 2	.001 .833 .004 .010
M_4d_	**VC** SI VO IN CO AR	**PR** BD VP MR FW AR	**WM** AR DS PS LN	**PS** CD SS CA	-	197.447	84	<.001	.973	0.966	.040 [.033 .047]	.033	58,846	59,088	M_4d_ vs. M_5e_	0.650	1	.394
M_5a_	**VC** SI VO IN CO	**VS** BD VP	**FR** MR FW	**WM** AR DS PS LN	**PS** CD SS CA	219.344	85	<.001	.968	0.960	.043 [.036 .050]	.036	58,867	59,104	M_5a_ vs. M_4d_ M_5a_ vs. M_5c_ M_5a_ vs. M_5d_ M_5a_ vs. M_5e_	21.897 12.175 19.630 21.247	1 1 1 2	<.001 .002 <.001 <.001
M_5b_	**VC** SI VO IN CO	**VS** BD VP	**FR** MR FW AR	**WM** DS PS LN	**PS** CD SS CA	317.222	86	<.001	0.945	0.933	.056 [.050 .063]	.064	58,965	59,197	M_5b_ vs. M_4d_ M_5b_ vs. M_5a_ M_5b_ vs. M_5c_ M_5b_ vs. M_5d_ M_5b_ vs. M_5e_	119.780 97.878 110.050 117.510 119.130	2 1 2 2 3	<.001 <.001 <.001 <.001 <.001
M_5c_	**VC** SI VO IN CO	**VS** BD VP	**FR** MR FW AR	**WM** AR DS PS LN	**PS** CD SS CA	207.169	84	<.001	.971	0.963	.041 [.034 .049]	.035	58,855	59,098	M_5c_ vs M_5e_	9.072	1	.001
M_5d_	**VC** SI VO IN CO	**VS** BD VP	**FR** MR FW	**WM** AR DS PS LN	**PS** CD SS CA	199.714	84	<.001	.972	0.965	.040 [.033 .047]	.034	58,848	59,091	M_5d_ vs M_5e_	1.617	1	.199
M_5e_	**VC** SI VO IN CO AR	**VS** BD VP	**FR** MR FW AR	**WM** AR DS PS LN	**PS** CD SS CA	198.097	83	<.001	.973	0.965	.040 [.033 .048]	.034	58,849	59,096				

*Note*: * = M4a was tested in the Technical and Interpretive Manual by [56] ([56]) and coincided with the four-factor exploratory model in our data.

**Table 6 jintelligence-12-00105-t006:** CFAs of the best-fitted models disaggregated by age group.

		Factors (Factor Loadings)					Fit Indices								
Model	Age Group	F1	F2	F3	F4	F5	χ2	df	*p*	CFI	TLI	RMSEA [90%CI]	SRMR	AIC	BIC
M_4a_	G1 (6–8)	**VC (.880)** SI (.678) VO (.743) IN (.605) CO (.551)	**PR (.790)** BD (.562) VP (.727) MR (.761) FW (.449)	**WM (.901)** AR (.776) DS (.809) PS (.557) LN (.757)	**PS (.594)** CD (.688) SS (.626) CA (.197)		120.124	86	.009	.964	0.956	.042 [.022 .059]	.048	15,502	15,669
	G2 (9–11)	**VC (.861)** SI (.831) VO (.816) IN (.792) CO (.667)	**PR (.866)** BD (.670) VP (.711) MR (.594) FW (.593)	**WM (.920)** AR (.702) DS (.761) PS (.590) LN (.683)	**PS (.458)** CD (.576) SS (.808) CA (.499)		105.730	86	.073	.985	0.982	.029 [.000 .047]	.040	18,168	18,344
	G3 (12–14)	**VC (.887)** SI (.880) VO (.815) IN (.779) CO (.702)	**PR (.804)** BD (.572) VP (.690) MR (.604) FW (.599)	**WM (.708)** AR (.687) DS (.753) PS (.686) LN (.775)	**PS (.556)** CD (.509) SS (.720) CA (.563)		155.831	86	<.001	.942	0.929	.060 [.045 .075]	.055	15,381	15,549
	G4 (15–16)	**VC (.774)** SI (.802) VO (.900) IN (.822) CO (.736)	**PR (.853)** BD (.696) VP (.634) MR (.763) FW (.781)	**WM (.891)** AR (.811) DS (.806) PS (.587) LN (.735)	**PS (.677)** CD (.675) SS (.731) CA (.629)		122.196	86	.006	.961	0.953	.055 [.030 .076]	.052	9570	9713
M_5a_	G1 (6–8)	**VC (.859)** SI (.680) VO (.759) IN (.587) CO (.549)	**VS (.814)** BD (.575) VP (.798)	**FR (.846)** MR (.791) FW (.489)	**WM (.854)** AR (.772) DS (.808) PS (.560) LN (.760)	**PS (.583**) CD (.664) SS (.650) CA (.201)	131.846	85	<.001	.951	0.939	.050 [.032 .066]	.051	15,516	15,687
	G2 (9–11)	**VC (.846)** SI (.833) VO (.818) IN (.788) CO (.668)	**VS (.826)** BD (.716) VP (.746)	**FR (.982)** MR (.583) FW (.588)	**WM (.910)** AR (.704) DS (.761) PS (.593) LN (.677)	**PS (.447)** CD (.571) SS (.814) CA (.498)	104.521	85	.074	.986	0.982	.029 [.000 .047]	.041	18,169	18,348
	G3 (12–14)	**VC (.838)** SI (.877) VO (.816) IN (.780) CO (.705)	**VS (.724)** BD (.619) VP (.799)	**FR (.983)** MR (.589) FW (.598)	**WM (.717)** AR (.690) DS (.752) PS (.688) LN (.771)	**PS (.524)** CD (.510) SS (.719) CA (.563)	151.314	85	<.001	.945	0.932	.059 [.043 .074]	.052	15,377	15,548
	G4 (15–16)	**VC (.758)** SI (.803) VO (.900) IN (.822) CO (.734)	**VS (.806)** BD (.823) VP (.690)	**FR (.859)** MR (.787) FW (.823)	**WM (.865)** AR (.810) DS (.806) PS (.589) LN (.736)	**PS (.678)** CD (.678) SS (.730) CA (.626)	112.915	85	.023	.970	0.963	.049 [.019 .071]	.052	9563	9709

## Data Availability

Data are unavailable due to ethical or privacy restrictions.

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
