# Peer review of "The Internal Structure of the WISC-V in Chile: Exploratory and Confirmatory Factor Analyses of the 15 Subtests"

_jintelligence, 2024, doi:10.3390/jintelligence12110105_

Round 1
Reviewer 1 Report
Comments and Suggestions for Authors
The manuscript brings a new perspective on the analyzed problem. 1. In order to improve its quality, some parts should be revised more thoroughly (language). 2. The abstract should include more results obtained.The manuscript brings a new perspective on the analyzed problem.
The manuscript brings a new perspective on the analyzed problem. 1. In order to improve its quality, some parts should be revised more thoroughly (language). 2. The abstract should include more results obtained.
Author Response
Comment 1:
The manuscript brings a new perspective on the analyzed problem.
In order to improve its quality, some parts should be revised more thoroughly (language).
Response 1:
Thank you for your comments.
- We revised the quality of the language
Comment 2:
The abstract should include more results obtained.
The manuscript brings a new perspective on the analyzed problem.
Response 2:
We included more information on the results about EFA, CFA, and internal consistency.
Reviewer 2 Report
Comments and Suggestions for Authors
This is an interesting report and it deserves publication. I do not have comments and suggestions for improving the Ms.
Author Response
Comment: This is an interesting report and it deserves publication. I do not have comments and suggestions for improving the Ms.
Response: Thank you very much.
Reviewer 3 Report
Comments and Suggestions for Authors
The Chilean study is well-described in terms of its proposal and the results obtained. It was a clever decision to extend the psychometric analysis of the WISC-V by comparing several factorial models, which were not presented in its initial study. Two minor observations I have is regarding the lack of description of the standardization sample used for the analysis. This information is presented in another document (written in Spanish), but it should be mentioned in this paper. For example, the sample was stratified by socioeconomic status, and it was based on municipal schools (286), subsidized private schools (245), and private schools (223). The sample was distributed in seven areas of the country: Arica and Parinacota, Antofagasta, Metropolitan Region, Maule, Biobío, Araucanía, and Los Lagos. The second observation concerns another, and also attractive, line of future research: testing the differentiation hypothesis by ability level (not just by age).
Author Response
Comment 1:
The Chilean study is well-described in terms of its proposal and the results obtained. It was a clever decision to extend the psychometric analysis of the WISC-V by comparing several factorial models, which were not presented in its initial study. Two minor observations I have is:
- regarding the lack of description of the standardization sample used for the analysis. This information is presented in another document (written in Spanish), but it should be mentioned in this paper. For example, the sample was stratified by socioeconomic status, and it was based on municipal schools (286), subsidized private schools (245), and private schools (223). The sample was distributed in seven areas of the country: Arica and Parinacota, Antofagasta, Metropolitan Region, Maule, Biobío, Araucanía, and Los Lagos.
Response 1: Thank you for your comments.We included more information on participant following the reviewer recommendation
Comment 2:
The second observation concerns another, and also attractive, line of future research: testing the differentiation hypothesis by ability level (not just by age).
Response 2:
We already had included this suggestion in lines 410-414, but added this idea in line 413.Reviewer 4 Report
Comments and Suggestions for Authors
Excellent and thorough job. Wonderful methodology, well-balanced and insightful coverage of the literature. Well written, well organized.
Author Response
Comment:
Excellent and thorough job. Wonderful methodology, well-balanced and insightful coverage of the literature.Well written, well organized
Response:
Thank you very much.Round 2
Reviewer 1 Report
Comments and Suggestions for Authors
I have no objections
Reviewer 3 Report
Comments and Suggestions for Authors
I agree with the publication of this paper.